# DArTSNP based genetic diversity analyses in cassava (*Manihote esculenta* [Cranz]) genotypes sourced from different regions revealed high level of diversity within population

**Neim Semman Abadura**[1,2]\*, **Abush Tesfaye Abebe**[3], **Ismail Yusuf Rabbi**[3], **Tewodros Mulualem Beyene**[1], **Wosene Gebresellassie Abtew**[2]

**1** Ethiopian Institute of Agricultural Research, Addis Ababa, Ethiopia, **2** Jimma University Colleges of Agriculture and Veterinary Medicine, Jimma, Ethiopia, **3** International Institute of Tropical Agriculture, Ibadan, Nigeria

\* neim2005eiar@gmail.com

## Abstract

Understanding the extent of genetic diversity is a pre-requisite in cassava breeding program due to its available broad genetic base of the crop and have great opportunity for its genetic improvement. This study was designed to assess the genetic diversity of 184 cassava germplasm sourced from International Institute of Tropical Agriculture (IITA) and previous collection of Jimma Agricultural Research Center (JARC) by using DArTSNPs markers. The data were subjected to imputation and filtering for minor allele frequency of 0.01, 0.95 major allele frequency using TASSEL and Beagle. The resultants 9,310 informative SNPs were retained and used to perform analysis of molecular variance (AMOVA), genetic diversity, population structure, and dissimilarity-based clustering of the tested cassava germplasm. The results of AMOVA revealed higher variation within (91.3%) than between (8.7%) the study populations. The high average PIC (0.44), expected heterozygosity (0.50), major allele (0.61) and minor allele (0.28) frequency showed the existence of high variation in the study populations. Population structure analysis grouped the panels into six structures with the existence of admixtures. Similarly, principal component analysis, factor analysis and cluster analysis apparently divided the panels into six clusters. Both the introduced and locally collected germplasm formed three clusters, each creating some mixes of genotypes, indicating that alleles sharing common ancestral background. The overall results, the studied genotypes showed significant variations, which can render opportunity for association mapping and technical conservation purposes.

## Introduction

Cassava (*Manihote esculenta* [Cranz]) is an ever-green semi-perennial root crop resilient to environmental hardship and grown in wider agro-ecologies [1]. Cassava is a diploid species with chromosome number of (2n = 18) originated in South America and spread to different

---

**Data availability statement:** All relevant data are within the paper and its Supporting Information files.

**Funding:** The author(s) received no specific funding for this work.

**Competing interests:** I have read the journal's policy and no competing interest for this manuscript.

continents and countries over a long period of time. It is highly heterozygous, primarily resulted from its cross-pollinated nature of the crop, when it propagated through botanical seeds [2]. However, the alternative propagating mechanism, clonally propagation, provided an opportunity for maintaining the diversity of the crop [3]. Worldwide, it is the fourth most important staple food next to rice, wheat and maize, and provide diets for more than a billion people [4]. Cassava is a very important root crop rich in carbohydrate and used for human consumption, animal feed and industrial applications. The crop has an edible tuberous root rich in starch and its leaf as a vegetable dish has become a popular food source in developing countries, widely in Sub-Saharan Africa (SSA) [5]. In most parts of the developing countries, besides for food purposes, cassava is used for the production of ethanol, alcohol, medicines and sources of income for smallholder communities [6]. The development of chips and pellets from cassava has played strategic significance in improving the marketing and exporting of cassava and promoting the expansion of cassava production worldwide [7].

In Ethiopia, cassava was first introduced in the middle of nineteenth century by some NGOs to combat drought in southern parts of the country primarily to fill the food shortage gap for subsistence farmers due to failure of other crops by drought stress [8,9]. Currently, the crop has extravagant gaining admiration as important sources of starch in Ethiopia, leading to its extensive cultivation in different areas of the country [10]. To gratify the growing demand by producers and users, cassava production needs to be extended to different parts of the country [11,12].

Diversity analysis is an important component of plant breeding and genetics. Several molecular marker technologies are available and have been used for diversity study, some of which includes Randomly Amplified Polymorphic DNA, allozymes, Single Nucleotide Polymorphism (SNPs) and simple sequence repeat (SSR). As a result of gel-based markers are limitations of genome coverage, labor intensive sequencing, requirement of prior sequenced reference, degree of resolution and lack of reliability, Diversity Array technology of Single Nucleotide Polymorphism (DArTSNPs) technology was developed by the Next Generation Sequencing Platform (NGS) for its reliability, high throughout put, sequence independent, the whole genome coverage and cost effectiveness [13]. In cassava panels, diversity study can be done using morphological markers, molecular markers or both. The former method is affected by environmental factors, whereas the advantage of the latter over the former is its consistency and independence of environmental factors in cassava characterization [14].

Genetic diversity and population structure assessment of 155 cassava genotypes were performed using 5,247 single nucleotide polymorphism markers by [15]. As a result, a total genetic variation of 66.0% within and 34.0% between sub-populations were quoted. These authors also reported five sub-populations in the population admixture analyses and three clusters in the cluster analysis that suggests the existence of high genetic diversity in the study germplasm. [15] conducted genomic analysis of 164 cassava collections from different cassava growing areas of Burkina Faso using selective 36 SNPs marker based on positions of the markers on the genome and reported average minor allele frequency, expected and observed heterozygosity and PIC of 0.06, 0.46, 0.58 and 0.36, respectively. On the other hand, [13] assessed 87 cassava accessions with filtered 10808 SNPs and 10521 SilicoDArT markers and reported informativeness of the markers with average PIC of 0.36 for SilicoDArT and 0.28 for SNP, marker, respectively. They also reported that SNP types were 50.60% transition and 49.40% transversion which were closely similar.

Hawassa and Jimma Agricultural Research Centers in Ethiopia have introduced several cassava germplasms for the past few years from the International Institute of Tropical Agriculture (IITA) and maintained them.

These cassava germplasms imported into the country at various times have not been assessed for their genetic diversity based on SNP markers that could have facilitated efficient

conservation of germplasm. Hence, there is limited information available on the extent of genetic diversity in the country that could have avoided conservation of bulk genotypes which makes the cassava genetic resource utilization inefficient. Hence, this study intends to assess the genetic diversity of cassava based on DArTSNP assay, which will have an immense contribution in the efficient exploitation of the cassava germplasm for future genetic improvement of the crop and conservation. Therefore, the main objective of this study was to assess the genetic diversity of cassava genotypes using DArTSNPs markers.

## Materials and methods

### Experimental materials

A total of 184 cassava genotypes were used in this study. Of these, 95 were introduced from the IITA cassava breeding program whereas the rest 89 were obtained from Jimma Agricultural Research Center (JARC) that were collected from the different cassava-growing areas of Ethiopia.

### Leaves sample preparation, DNA extraction and genotyping

Leaf samples were collected from 15 days young healthy cassava seedlings into microplate using leaf puncher carefully with regular cleaning of the puncher with alcohol after sampling each genotype to prevent contamination. During samples collection, the plate with samples was stored in cold ice box to retain normal leaf tissue. In the laboratory, the well plate accommodating leaf tissues were stored at -80°C until freeze-dried and shipped for sequencing. The freeze-dried leaf samples were sent to Intertek Australia for DNA extraction and KASP genotyping (using QC and trait-linked markers). It was subsequently forwarded to DArT for DArTseqLD analysis. The freeze-dried leaf tissues of each genotype were ground in an Eppendorf tube added with a metal bead using geno-grinder. Then, following procedures developed by [16], total genomic DNA was extracted. The DNA quality and quantity were checked using both spectrophotometer at absorbance ratio of A260/A280 using reference range of 1.8 and 2.0, and gel electrophoresis using 0.8% agarose gel. Then, using complexity reduction restriction enzyme, i.e., the type II ApeKI, which is target-full cutter that recognize five based pair sequence (GCWGC), where "W" can be either A or T and the genome was cut into fragments. Fragments of DNA were ligated using enzyme ligase and barcoded for matching sequence of unknown DNA sequence of genotypes with known, by comparing with reference libraries.

### DNA sequencing and filtering

Extracted DNA was sequenced using the Illumina HiSeq 2500. The genotyped data was obtained in raw hapmap format. The marker quality evaluation was made where individual marker-related analysis was performed as suggested by Uba *et al.* [17]. Data imputation followed by filtering was done using TASSEL ver.5.2 [18]. Accordingly, a total of 12840 SNPs of DArTseq derived data were imputed by mean and further filtered for non-informative data with minor allele frequency of 0.01 and maximum allele frequency of 0.95. As a result, 9310 SNPs were retained for further diversity and related analysis.

## Data analysis

### Assessment of AMOVA and frequency-based genetic diversity

A total of 12840 SNPs was obtained from sequencing 184 cassava genotypes. The missing data filtering by site followed by imputation was done using beagle. After processing the total SNPs for filtering and imputation, the whole genotypes with 9310 SNPs were generated and used for

diversity analysis. Analysis of molecular variance (AMOVA) was conducted using 'adegenet' and 'poppr.amova' function of r studio ver.4.3.3. Basic diversity indices, such as polymorphic information content (PIC), observed and expected heterozygosity, number of alleles per locus, major allele frequency, breeding coefficient and Shannon index (I) were analyzed using power marker software and confirmed with GenAlex software ver. 6.5 [19].

PIC value was calculated as

$$PIC = \sum P_i^2 - \sum 2P_i^2$$

Estimates of the intra-genetic variation and genetic differentiation among populations were determined using GenAlex software in Microsoft excel. It was calculated using the formula.

$$PhiPT = AP/(WP + AP)$$

Where: AP = Estimate of variance among population, WP = Estimate of variance within population

Additionally, pairwise Nei genetic distance between population and within individual genotypes and allele frequency were analyzed using power marker software [20].

The number of clusters, genetic diversity or fixation index within cluster and between clusters, as well as expected heterozygosity between individuals within the population was determined using population structure software.

## Genotypes relatedness analysis

Cluster analysis of the study germplasm was carried out using R software version 4.3.3. The analysis was conducted using "factoextra" package and "hclust" r-function and also confirmed using DARwin user-friendly software using neighbor-joining method and dissimilarity matrix based factorial analysis [21]. The optimum number of clustering was determined using elbow method of fviz, nbcluster function ("Fig 5"), which gave a clue for determining cluster number.

The dissimilarity-based genotypes grouping was done using population structure in r software using LEA packages. Similarly, appropriate number of clusters was determined using Bayesian information content (BIC) in R using *genind* and *find.custers* functions. The principal component analysis of 184 cassava germplasm with 9,310 DArTSNPs markers was conducted using r-software of 'princomp' package.

## Result

### Analysis of Molecular variance (AMOVA)

Out of the generated 12,840 DArTSNPs data of 184 cassava genotypes subjected to filtering, a total of 9310 (72.51%) SNP markers were retained and distributed over the 18 chromosomes with an average of 517.22 for analysis (S1 Table). The result of AMOVA indicated that a higher percentage variation was observed within populations 91.33% (Phi = 0.913) than among populations 8.67% (Phi = 0.087) ("Table 1"). Additionally, the frequency of alleles' mutation or genomic DNA mutation, i.e., the percentage transitions (Ti) and transversion (Tv) were 55.9% and 44.1%, respectively ("Fig 1"). As indicated in "Table 1", the population variance (Phi) was 0.91, which indicated that there was highly significant variation across the studied panels.

The density of single nucleotide polymorphism (SNP) distribution along the whole genome of cassava chromosome was plotted in 1Mb (mega base) window size ("Fig 2"). Despite varying densities, the SNPs were distributed across the whole genome. The largest number

Table 1. AMOVA among and within 185 cassava genotypes using SNP marker.

| Sources of variation | DF | SS | MS | % variance | Phi | P-value |
|---|---|---|---|---|---|---|
| Among population | 1 | 40558.84 | 40558.84 | 8.67 | 0.087 | 0.0064 |
| Within population | 182 | 759459.8 | 4172.86 | 91.33 | 0.913 | |
| Total | 183 | 800018.6 | 4371.69 | 100 | | |

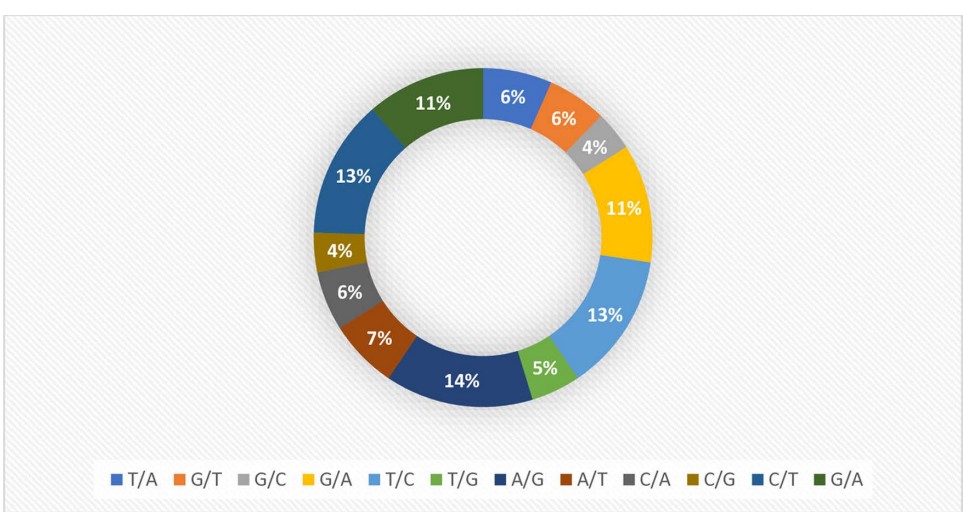

Fig 1. Percentage distribution of purine and pyrimidine in 9310 SNPs.

Fig 2. SNP density plot across cassava genome of studied panel.

of SNPs density (10.2%) was found on chromosome one, followed by chromosomes 14 and four, whereas chromosomes 13 and 7 had less density of SNPs ("Fig 2 and S1 Fig"). Moderate numbers of SNPs were observed on chromosomes 3, 10 and 6, respectively.

Also, the percentage Shannon diversity information revealed the presence of high within-population variation of 62%, whereas the between populations variation was 38%, respectively ("Fig. 3").

Percentages of Shannon Information (sH)

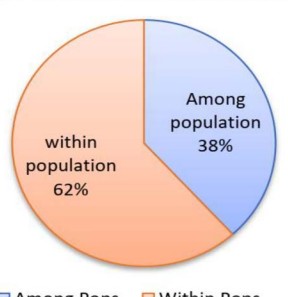

Fig 3.  percentage of Shannon diversity index among and within populations.

## Analysis of population statistics of studied panels

Diversity parameters such as average, maximum and minimum numbers of markers per locus, polymorphic information content (PIC), expected and observed heterozygosity, pairwise genetic distance within genotypes, between the group of populations and within individuals in the group, major and minor allele frequency were computed and presented in ("Table 2"). Accordingly, the maximum and minimum polymorphic information content (PIC) values were 0.70 and 0.03, respectively, with mean values of 0.44. Similarly, observed heterozygosity varied from 0.13 to 0.46 with a mean of 0.25. The Shannon diversity index among and within populations were 1.95 (38%) and 3.1 (62%).

The genetic distance between each of the genotypes based on the Nei distance method also depicted the presence of high diversity within the study panels that ranged from 0.237 between the genotypes G171 and G106, and 0.64 between G3 and G155 ("S2 Table").

On the other hand, the gene distance value among germplasm detected the presence of considerable genetic diversity. For these retained SNP markers, the maximum, minimum and average gene diversity (He) was 0.7491, 0.0320 and 0.505, respectively. On the other hand, the maximum, minimum and average major allele frequencies recorded were 0.9837, 0.6115 and 0.2663, respectively ("Table 2"). The coefficient of inbreeding of the panels also showed moderate variability of gene diversity with the mean, maximum and minimum values of the population 0.6797, 0.836 and 0.407, respectively.

Grouping of genotypes, analysis of expected heterozygosity, and fixation index of the studied cassava genotypes were also performed during population structure analysis using STRUCTURE software and ultimately, the genotypes were categorized into six groups. The divergence analysis between the clusters has also displayed the presence of variability between these clusters. The highest fixation index (Fst) of 0.2784 was found between cluster-V and cluster-II, followed by between cluster-V and cluster-I (0.2303), whereas the lowest Fts value of 0.0747 was found between cluster-VI and cluster-III ("Table 3").

## Population structure and admixture analysis

The population structure of the study cassava panels was assessed using principal component, cluster and factorial analyses based on a dissimilarity matrix. The results of the population structure analysis of the studied panels are presented in "Fig 4". As a result, the populations were grouped into six structures with appropriate delta K-values of six. However, admixture was observed in all the structured populations.

**Table 2. Mean, maximum and minimum summary of population genetic parameters 184 cassava germplasm using 7864DArTSNP markers.**

| Parameters | Mean | Maximum | Minimum |
|---|---|---|---|
| Major allele frequency | 0.61 | 0.98 | 0.26 |
| Minor allele frequency | 0.28 | 0.52 | 0.05 |
| PIC | 0.44 | 0.70 | 0.03 |
| Observed heterozygosity (Ho) | 0.25 | 0.47 | 0.13 |
| Gene diversity (He) | 0.50 | 0.75 | 0.03 |
| Number of allele/locus | 3.00 | 4.00 | 2.00 |
| Inbreeding coefficient (f) | 0.68 | 0.84 | 0.41 |

**Table 3. Divergence (fixation index) among clustered populations based on net nucleotide distances.**

| | Cluster I | Cluster II | Cluster-III | Cluster-IV | Cluster-V | Cluster-VI |
|---|---|---|---|---|---|---|
| Cluster I | – | | | | | |
| Cluster II | 0.17 | – | | | | |
| Cluster-III | 0.08 | 0.19 | – | | | |
| Cluster-IV | 0.11 | 0.16 | 0.1 | – | | |
| Cluster-V | **0.23** | **0.28** | **0.22** | **0.19** | – | |
| Cluster-VI | 0.10 | 0.18 | 0.07 | 0.08 | 0.19 | – |

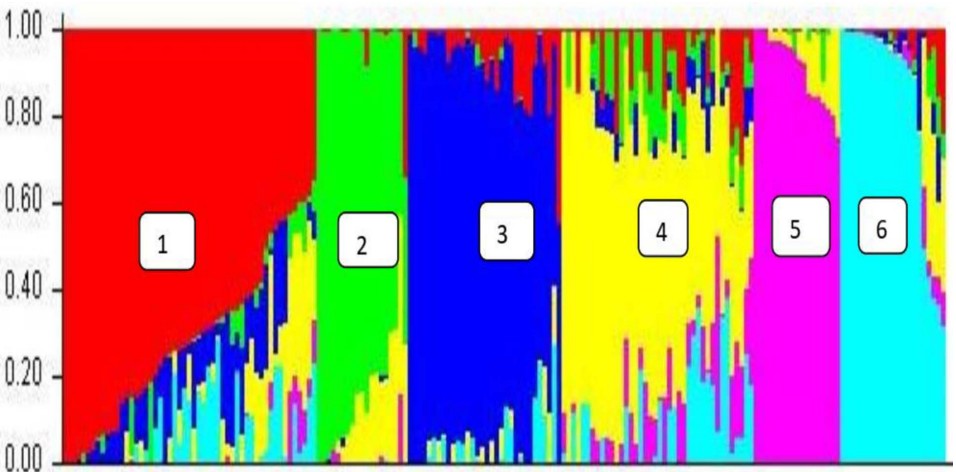

**Fig 4. Population structure and admixture analysis of individuals in the panel.**

## Cluster analysis

Similar to the results of the population structure, the cluster analysis using Bayesian information content (BIC) in r-package ("Fig 6") as well as factorial analysis using the neighbor-joining method and dissimilarity matrix using DARwin software ("S2 Fig") categorized the study germplasm into six groups. Accordingly, genotypes introduced from IITA formed cluster I, II and III, whereas the genotypes sourced from Ethiopia that were previously collected and maintained at JARC formed clusters IV, V and VI.

The fourth cluster comprised of the largest, number (57) of cassava genotypes, followed by the third cluster (47), while the first, second and sixth clusters contained nearly equal numbers, i.e., 18, 18 and 17 of cassava genotypes, respectively.

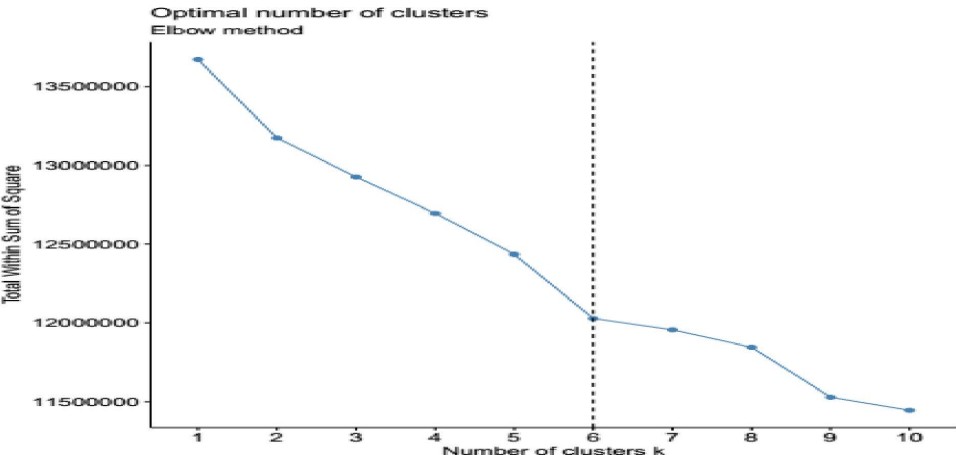

**Fig 5. Determining number of clusters using elbow method.**

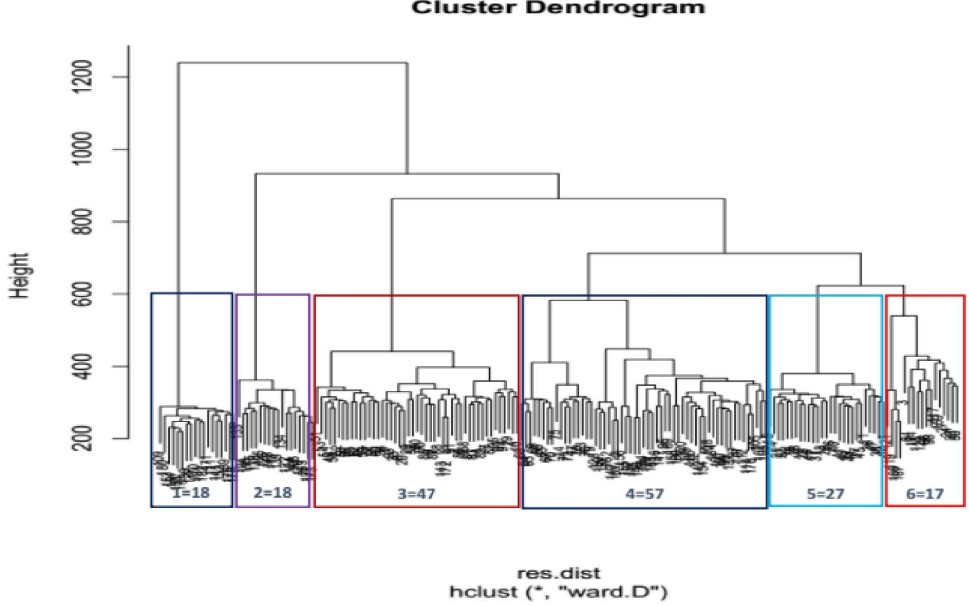

**Fig 6. Dendrogram representing groups of related genotypes.**

## Principal component analysis

The result of principal component analysis displayed the presence of high genetic diversity in the study germplasm, where the first and second principal components (PCs) contributed to 18.2% and 14.8% of the total variation, respectively. The first two PCs cumulatively explained was 33.1% of the total variation ("Fig 7 and Fig 8").

## Discussion

The genetic variability study in cassava is very intrinsic as the crop is highly cross pollinated and heterozygous in nature. This nature of the crop provides the opportunity to identify

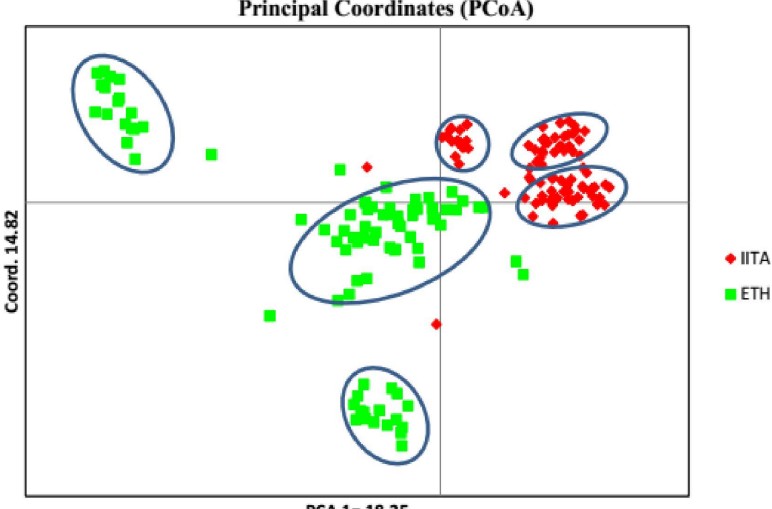

**Fig 7. Principal coordinate analysis of 184 cassava based on DArTSNPs.**

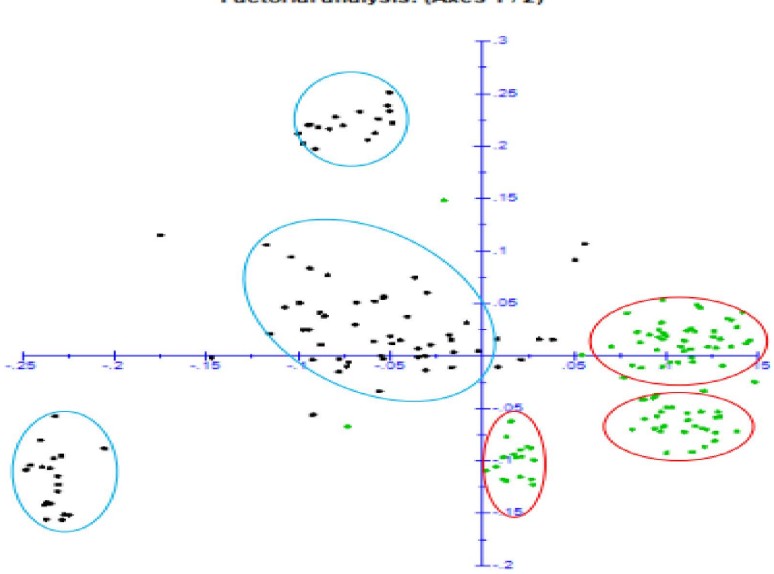

**Fig 8. Similarity-based factorial analysis of 184 cassava germplasm.**

parental genotypes with traits of interest, whereas the vegetative propagation mechanism of the crop provides sustainable preservation of its diversity [22]. The genetic diversity assessment of this crop also renders researchers to develop breeding strategies that ensures continuous genetic improvement of the crop and effective utilization of genetic resources in conservation strategies.

All of the genetic diversity parameters considered in this study revealed the high genetic diversity of the studied germplasm. These parameters were highly informative and consistent with the values reported by other authors. The polymorphic information content (PIC) that evaluates the informativeness of molecular data was 0.762 with a mean of 0.44, which was far

higher than the PIC values previously reported by [13]; [23] and [24] which was 0.2-0.5, with means of 0.36, 0286 and 0.6057 respectively. In previous cassava genetic diversity study using SNP markers, on 102 white and yellow accession, [5] reported a PIC value 0.17, which was far lower than the PIC values in this study.

The DNA sequence mutation, i.e., transition (Ti) and transversion (Tr) are the other factors contributing to the genetic variability in crop plants [21]. The types of SNPs mutation analysis in this study revealed more transition (54.9%) that transversion (45.1%) similar to the finding of [25] who reported transition SNPs 60.76% (4,470 SNPs) and transversion SNPs of 39.24%, (2887 SNPs) in cassava. [4] reported nearly similar transition (50.6%) and transversion (49.4%) in this study. This might explain that transition mutation is more likely to occurs than transversion due to similar ring base mutation more likely occur than distinct ring bases replacement [25]. This agrees with the result of Sesay et al. [26] who reported higher 60.8% transitions that transversion 39.2% transversion. Similarly, out of the 698,347 variants, [27] reported 54% transition (C/T or A/G) and 42% transversion (A/C, A/T, C/G or G/T) mutations, which showed more frequency of transition mutation than transversion. Regarding the SNP distribution and its density along the cassava genome, the highest density of SNPs was found on chromosome 1 which is in agreement with the finding of [26].

The average values of observed heterozygosity 0.25 that ranged from 0.13-0.5 and gene diversity (He) 0.5 found in this study revealed the presence of high diversity across the studied panels was in concordance with the finding of Esuma et al. [28] who reported expected heterozygosity of 0.56. The gene diversity of 0.63 and observed heterozygosity (0.57) in 1401 cassava cultivars and landraces collected from seven Sub-Saharan countries assessed using microsatellite markers by [28]; and average He of 0.72 and 0.69 of Ho portrayed by Tiago et al. [29] are higher than the results in this study. On the other hand, Goncalves et al. [30] assessed 51 cassava accessions of Minas Gerais state of Brazil and reported observed heterozygosity of 0.65indicating moderate genetic variance with gene diversity of 0.48 and PIC of 0.4. However, in the current study, the observed heterozygosity (0.25) was less than the expected heterozygosity (0.5) departing from Hardy-Weinberg equilibrium which is in agreement with the report of [31,32] who reported Ho (0.34) and He (0.58) which might possibly be attributed to inbreeding.

Population differentiation index is the fundamental indicator of population phylogeny and is statistical measure of genetic differentiation of the populations under consideration [33]. The population differentiation (Phi)value (0.701) obtained in this study was also in consistent with the other parameters that validated the presence of high genetic variability in the study population. These diversities of the cassava populations partitioned into among the populations and within individual of the population. Accordingly, within-population diversity accompanies the lion share of 91.3% whereas among population variance was 8.7% respectively. Analysis of molecular variance in this study revealed high variability within individuals in the population; and low variability between populations respectively. This result was in line with the findings of [34] who reported 91% and 9% variance within individual and among populations. Adjebeng-Danquah et al. [35] assessed 89 cassava accessions using SSR markers in Ghana and reported 97% within-group variation and 3% between group variations. Similarly, in the genetic diversity study of landraces cassava in Ghana using SNP markers, Prempeh et al. [24] reported 99% within and only 1% between group variation.

## Genetic relatedness and differentiation analysis

Principal coordinate, cluster and population structure are some of the analyses that indicate similarity of individuals within populations based on some of the most important traits, where genotypes with related traits are grouped together forming clusters.

Genetic relationship of 184 cassava germplasm was assessed using principal coordinate analysis (PCA), population structure, cluster analysis and factorial analysis based on 9310 DArTSNPs markers. The population structure analysis of the study panel revealed six structures forming distinct groups ("Fig 3"). However, high degree of admixture was observed across all the six structures. This is possibly due to allele sharing. Assessment of fixation index performed within and among clustered population achieved above were greater than the result quoted by [1] < 15% for one hundred cassava landraces collected from different agro-ecology of Burundi. The authors also reported the Fst, He and Ho within the individuals of the population ranged from 0.57-0.6, 0.25-0.27 and 0.25-0.27 respectively.

Other similarity-based analyses supported the grouping of the studied populations into six groups with the existence of admixtures of the different populations.

The dendrogram plotted on the basis of neighbor joining method of the studied populations revealed six clusters. The results of population structure and principal coordinate analyses categorized the germplasm introduced from IITA (population I) into the first three clusters (1, 2 and 3) whereas the genotypes sourced from Ethiopia (population II) also were divided into three distinct cluster. However, some sorts of admixtures were observed. This is well depicted in the result from similarity based clustering ("Appendix fig 2"), principal coordinate analysis (PCA) ("Fig 5") and factorial analysis ("Fig 6"). In the PCA analysis, the first and the second PC explained 18.25 and 14.82% of total variation, respectively with the first two PCs cumulative contribution of 33.1% of the total variation among 184 cassava germplasm. This finding was in agreement with the result of [29] who reported 13.4% and 6.08% respective contribution of the first and the second PCs to the total variability found in 157 cassava landraces. The formation of admixture in population structures of this study indicated the relatedness of individuals within the population, which might be due to sharing common ancestor. This was in accordance with the results of [29] who conducted study with six cassava populations of 157 ethno-varieties based on their sources of origin. The similarity-based analysis of the six populations generated two clusters and the authors reported that similar cassava ethno-varieties are being cultivated in different regions. In agreement with the above finding [35] reported clustering of 122 traditional sweet cassava cultivars into six groups, indicating the possibility of alleles shared within the genotypes.

## Conclusion

The genetic diversity parameters based on the DArTSNPs markers considered in this study were informative and indicated the studied panels were divergent. The similarity-based analyses such as cluster, principal component, population structure and factorial analyses categorized 184 cassava genotypes into six groups with high values of fixation index. Between different cassava populations, admixtures occurred, which might be as a result of germplasm exchange during introduction and common ancestral. From the result, it can be inferred that association mapping can provide putative gene/s and provide the breeders with high variation for the selection of parental line with traits of interest for cross breeding. Also, it can provide the way forward for researchers to develop strategic conservation methods.

## Supporting information

**S1 Fig.  Percentage of SNPs distribution along the 18 cassava chromosomes.**
(TIF)

**S2 Fig. Similarity based grouping of 1he 184 cassava genotypes using SNPs dataset.**
(TIF)

**S1 Table. Filtered DArTSNPs_9319 SNPS dataset.**
(XLSX)

**S2 Table. Summary of Nei genetic distance.**
(XLSX)

## Acknowledgement

We are grateful to express our appreciation to all who contributed their fingerprint in success of this study.

## Author contributions

**Conceptualization:** Neim Semman Abadura, Abush Tesfaye Abebe, Ismail Yusuf Rabbi, Tewodros Mulualem Beyene.

**Data curation:** Neim Semman Abadura.

**Formal analysis:** Neim Semman Abadura.

**Funding acquisition:** Ismail Yusuf Rabbi.

**Investigation:** Neim Semman Abadura, Ismail Yusuf Rabbi.

**Methodology:** Neim Semman Abadura, Abush Tesfaye Abebe, Ismail Yusuf Rabbi, Tewodros Mulualem Beyene, Wosene Gebresellassie Abtew.

**Supervision:** Abush Tesfaye Abebe, Ismail Yusuf Rabbi, Tewodros Mulualem Beyene, Wosene Gebresellassie Abtew.

**Validation:** Ismail Yusuf Rabbi.

**Visualization:** Ismail Yusuf Rabbi, Wosene Gebresellassie Abtew.

**Writing – original draft:** Neim Semman Abadura.

**Writing – review & editing:** Neim Semman Abadura, Abush Tesfaye Abebe, Ismail Yusuf Rabbi, Wosene Gebresellassie Abtew.

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
