## [Decision Letter · Decision Letter 0]

13 Nov 2024

PONE-D-24-29508DArTSNPbased genetic diversity analyses in cassava (Manihot esculenta) genotypes sourced from different regions revealed high level of diversity within populationPLOS ONE

Dear Dr. Abadura,

Thank you for submitting your manuscript to PLOS ONE. I have reviewed the manuscript and the reviewer comments. However, after careful consideration, we feel that it has scientific merit but does not fully meet PLOS ONE’s publication criteria as it currently stands. Therefore, we invite you to submit a revised version of the manuscript that addresses the points raised during the review process.

Be sure to address the following changes:

 English editing is needed due to several grammatical errors. For example, there is too much incorrect usage of abbreviations; some unnecessary repetition and formatting must be addressed. Abbreviations must be introduced first and then used after that.The figures need to be arranged correctly in numerical order. See Figure 1. Figure 1 is listed twice. I see no mention of Figure 5 in the results section.The methods section of the manuscript also requires major revision. The age of the plants is not described properly. How young were they? Were they all the same age?Table 1 is unnecessary since the authors already mentioned the information in the text.Several times in the manuscript, the authors mention software programs, but the author or version of the software programs is not referenced. One example is Darwin, and others include Power Maker and GeneAlex. Also, sometimes the version of R is listed, but other times it is not. These are just some that I noticed throughout the manuscript.Where was the sequencing done?There is no mention of the supplier of the ligation enzyme reagent. Was a kit used?The results show an inconsistency in the use of significant figures. Be consistent. Are you using two or three significant figures? It can be confusing if something is listed in the table using two significant figures and is described with four significant figures in the text.In Figure 3, use the term Shannon diversity index as you have used throughout the manuscript.Figure 3 should be Figure 4. Label this figure with the numbers 1-6 to show the six panels.Pay attention to the numbering of the figure here.

We look forward to receiving your revised manuscript.

Kind regards,

Angela T. Alleyne, Ph.D

Academic Editor

PLOS ONE

Journal Requirements:

Reviewers' comments:

Reviewer's Responses to Questions

**Comments to the Author**

1. Is the manuscript technically sound, and do the data support the conclusions?

Reviewer #1: Yes

2. Has the statistical analysis been performed appropriately and rigorously? 

Reviewer #1: Yes

3. Have the authors made all data underlying the findings in their manuscript fully available?

Reviewer #1: Yes

4. Is the manuscript presented in an intelligible fashion and written in standard English?

Reviewer #1: Yes

5. Review Comments to the Author

Reviewer #1: I thoroughly examine the Manuscript ID: PONE-D-24-29508, entitled “DArTSNPs based genetic diversity analyses in cassava (Manihote esculenta [Cranz]) genotypes sourced from different regions revealed high level of diversity within population” which was submitted for Plos One journal for publication.

The manuscript was a well written article in all respects. The abstract, introduction, methodological aspects, presentation of results and discussion are sound and it is written in good language. Excellent work has been done to identify collection of casava accessions using molecular analysis in the country. I feel the subject is worthy enough for publication. Authors have clearly provided justification of genetic diversity and used appropriate statistical tools for this study for analysis. The conclusion is also sounds great. I did not find any major corrections to be made except some typographical errors as indicated below.

Abstract:

• AMOVA revealed higher variation within (91.3%) and between (8.7%) the study populations.

Text

• In text the reference should start with [1], not with the [13]. It needs to be in chronological order….please double check the references in text and reference list.

6. PLOS authors have the option to publish the peer review history of their article (what does this mean? ). If published, this will include your full peer review and any attached files.

**Do you want your identity to be public for this peer review?** For information about this choice, including consent withdrawal, please see our Privacy Policy .

Reviewer #1: **Yes: ** Abe Gerrano

---

## [Author Response · Author response to Decision Letter 0]

1 Dec 2024

First of all, I want to appreciate you for your time devotion and providing to me constructive comments and suggestion. I positively accepted your comments and my responses are as follow.

1. I was rigorously revised for grammatical improvement, first introducing of abbreviation, and correcting appropriate formatting.

2. Concerning figure numbering and ordering as well, I have corrected figure citation in body. I highlighted in yellow color.

3. The age of the seedlings from which sampling carried out has been described

4. Under result part, by accepting suggestion, table 1 has been deleted. However, I think it does not matter if available because table summarize and we describe what we summarized. Based on your comments, where sequencing done, it has been well described.

5. For the software used, which its version was not mentioned has been revised and corrected

6. As far as ordering figures and table ordering, I have corrected it according to its proper orders

Generally, I highlighted all correction undertaken in accordance with the comment, suggestion and format of PLOS ONE.

Thank you

---

## [Editor Report · Decision Letter 1]

24 Jan 2025

DArTSNPbased genetic diversity analyses in cassava (Manihot esculenta) genotypes sourced from different regions revealed high level of diversity within population

PONE-D-24-29508R1

Dear Dr. Abadura,

We’re pleased to inform you that your manuscript has been judged scientifically suitable for publication and will be formally accepted for publication once it meets all outstanding technical requirements.

Kind regards,

Angela T. Alleyne, Ph.D

Academic Editor

PLOS ONE
---

## [Editor Report · Acceptance letter]

PONE-D-24-29508R1

PLOS ONE

Dear Dr. Abadura,

I'm pleased to inform you that your manuscript has been deemed suitable for publication in PLOS ONE. Congratulations! Your manuscript is now being handed over to our production team.

Kind regards,

on behalf of

Dr. Angela T. Alleyne

Academic Editor

PLOS ONE